# Comparing the Short-Term Outcome after Polytrauma and Proximal Femur Fracture in Geriatric Patients

**DOI:** 10.3390/jcm10061287

**Published:** 2021-03-20

**Authors:** Andreas Gather, Tomoko Tajima-Schneider, Paul A. Grützner, Matthias Münzberg

**Affiliations:** Clinic for Trauma and Orthopaedic Surgery, BG Trauma Center Ludwigshafen, University of Heidelberg, Ludwig-Guttmann-Str. 13, 67071 Ludwigshafen on the Rhine, Germany; andreas.gather@bgu-ludwigshafen.de (A.G.); tajimaschneider.tomoko@gmail.com (T.T.-S.); paul.gruetzner@bgu-ludwigshafen.de (P.A.G.)

**Keywords:** severe injured patients, outcome, geriatric, proximal femur fracture

## Abstract

Because of demographic change, geriatric patients are becoming a major challenge for traumatology. Multiple trauma patients and patients with proximal femoral fractures are important groups of patients in geriatric traumatology. This retrospective study compares two patient groups with different severities of injuries, and analyzes their patient characteristics and short-term outcomes, focusing on functionality upon discharge. The investigation aims to present the characterizing features of both patient groups, and to identify the potential risk factors for early functionality after trauma. The patient collective comprises two patient groups: a polytrauma group with 91 patients, and a femoral fracture group with 132 patients. Under the control of potential influencing factors, the present study showed no significant influence of belonging to either of the patient groups (multiple trauma or proximal femoral fracture) on the mobility status at discharge. Age, known dementia, pre-clinical intubation, and the lowest Hb value were identified as significant influencing factors. Despite their old age and vulnerability, the majority of geriatric patients survive accidents. Further prospective investigations concerning the maintenance or restoration of functionality after an accident are therefore desirable.

## 1. Introduction

Due to demographic changes, the number of geriatric trauma patients is becoming a major challenge for hospitals and society [1]. A retrospective evaluation of the TraumaRegister DGU^®^ over 20 years has already shown that the proportion of patients over 60 has increased continuously, from almost 15% in 1993 to over 30% in 2012. The Federal Statistical Office of Germany’s official population projection also assumes an increase in the proportion of the population over 64 years of age from 21% in 2013 to 33% in 2060 [2]. In the foreseeable future, a further increase in elderly patients in traumatology can be expected. Geriatric patients are defined by a geriatric-typical multimorbidity and an older age (mostly 70 years or older). Patients aged 80 and over are considered geriatric due to their age-typical increased vulnerability, such as the occurrence of complications and the risk of a loss of autonomy. The physiological aging process and the reduced physiological reserve capacity, which are associated with a limited ability to compensate for internal or external changes in the environment adequately, make supply more difficult [3]. It is, therefore, essential to pay particular attention to the outcomes of these patients. The question of which therapy should be initiated in a severely injured geriatric patient has not been fully clarified. If there is a patient decree with a limited therapeutic will, the patient already gives the direction. Such a question would rarely arise in the case of a proximal femur fracture. Therefore, it is important to research whether the outcomes of these two injury entities are so significantly different. Many of the outcome studies examine mortality and potential influencing factors. The hospital mortality of geriatric multiple trauma patients without traumatic brain injury varies between 14% and 29%, and in patients with traumatic brain injury, it varies between 57% and 64% [1,4]. So far, only a few studies have compared the predictive power of different scoring systems in geriatric patients concerning their prognosis after multiple trauma [5,6]. The question of whether a geriatric patient survives an accident or not is undoubtedly clinically relevant. After an accident, however, medical care’s actual goal is not only to ensure pure survival but also to ensure functional restoration that allows the patient to return to their previous life. Depending on the trauma mechanism, between 50% and 85% of the patients in one study showed good functionality after a severe injury [7]. In conclusion, it can be stated that, compared to the many mortality studies, and despite the clinical and socio-economic relevance, there are significantly fewer studies that examined the functionality and outcome of polytraumatized geriatric patients. In contrast, there are numerous studies on the outcome of proximal femoral fractures. A proximal femur fracture as a single injury and a multiple trauma are very different in terms of the severity of the injury. Based on previous studies, the hospital mortality in multiple trauma patients tends to be higher than in patients with a proximal femoral fracture. Regarding the functionality after the trauma, however, it is unclear whether there is a significant difference between multiple trauma patients and patients with a proximal femoral fracture.

On the one hand, there are few studies on the short-term functionality of geriatric patients after a multiple trauma; on the other hand, it can be assumed from studies in geriatric patients with a proximal femur fracture that patient-related factors influence the outcome more than the severity of the injury. Even if the groups seem different in terms of injury severity, the comparison is important for everyday clinical practice. The functionality at the end of the inpatient stay determines the form of further treatment, but does not allow any conclusions to be drawn about the quality of life months after the injury. This study presents the functional outcome at the end of the inpatient stay, and compares both patient groups. In summary, this retrospective study was realized in order to compare geriatric multiple trauma patients with geriatric patients with a proximal femur fracture. 

## 2. Materials and Methods

### 2.1. Collection of the Patient Data

The multiple trauma group were patients with an Injury Severity Score (ISS) ≥ 16, who were admitted to a level 1 trauma center in Germany from 1 January 2010 to 31 December 2015, and who were 70 years or older at the time of admission. Patients with a traumatic brain injury with an AIS head ≥ 5 were excluded here, as this is most likely to be a traumatic brain injury as the leading injury. According to these criteria, 135 patients were included in this study as the multiple trauma group, but 44 of them had to be excluded due to a later-noticed severe traumatic brain injury. During the same observation period, a total of 822 patients diagnosed with a ‘proximal femur fracture’ (femoral neck fracture, pertrochanteric femur fracture, or subtrochanteric femur fracture), who were 70 years or older at the time of admission, were admitted to the BG Trauma Center Ludwigshafen. Patients with a femoral shaft fracture or a distal femoral fracture, and patients with multiple injuries were excluded. From this group of patients, 22 patients per calendar year were randomly selected in order to achieve a similar group size. The randomization was carried out as follows: a random number was generated and assigned to each patient per calendar year using an Excel function. The patients were then sorted according to this random number and the first 22 patients were selected, giving a total of 132 patients included in the femoral fracture group of this study. 

This study was approved by the ethics committee of the Rhineland-Palatinate State Medical Association (2020-15129-retrospective).

### 2.2. Collected Parameters

The parameters shown in Table 1 were recorded for both patient groups. On the one hand, these parameters were selected in order to show the descriptive characteristics of the respective patient group and, on the other hand, to be included as potential influencing factors (confounders) in the multiple regression analysis.

### 2.3. Statistical Methods

The statistical evaluation was performed with the statistics program SPSS for Windows^®^, version 25.0.

The presentation of the group characteristics is descriptive. A mean ± standard deviation was given for the quantitative parameters as a measure for normally distributed data, and a median and the 25%/75% quantile was given for the non-normally distributed data. The absolute and relative frequency distribution was shown for the qualitative parameters. A Kolmogorov–Smirnov test or a Shapiro–Wilk test was used to check a parameter for a normal distribution. The differences between the two patient groups concerning the mean, the central tendency, or the frequency distribution were examined using different statistical methods, depending on the parameter’s scale level. All of the statistical test procedures used were based on a significance level of 5%.

A multiple linear regression analysis was conducted in order to examine the significance of the factors influencing the mobility status at discharge for the patients who survived their accident.

## 3. Results

### 3.1. Patient Characteristics

The median patient age at admission was 76 years for the multiple trauma group (25% quantile: 73 years; 75% quantile: 79 years), and was 81 years for the femoral fracture group (25% quantile: 78 years; 75% quantile: 85.8 years). On average, the patients in the femoral fracture group were 5 years older. This age difference is statistically significant (*p* < 0.001).

The gender distribution in the two groups was very different, and this difference was statistically significant (*p* < 0.001). In the multiple trauma group, the proportion of male patients was 69.2%. In contrast, the proportion of male patients in the femoral fracture group was only 28.0%.

In both groups, the median number of secondary diagnoses was four. There was no statistically significant difference between the groups (*p* = 0.944).

In order to record the patients’ cognitive impairment on admission, the patients were examined to determine whether dementia was already known as a secondary diagnosis at admission. The proportion of patients with known dementia was 19.7% in the femoral fracture group, which was significantly higher than in the multiple trauma group, which had 5.5% (*p* = 0.003).

Two-thirds of the multiple trauma group and more than half of the femoral fracture group took neither platelet aggregation inhibitors nor anticoagulants. With 25.8%, significantly more patients in the femoral fracture group took ASA than in the multiple trauma group with 11.1%.

Table 2 shows the patients’ pre-clinical data related to intubation, resuscitation, and catecholamines. 

Based on the trauma-related data, it can be seen that the median of the ISS values of the multiple trauma group was 29. The trauma causes in the multiple trauma group were falls from a height (falls from a height of over 1.5 m to 3 m: 16.5%; over 3 m to 6 m: 14.3%) and traffic-accidents. In contrast, 81% of the patients suffered a femur fracture due to falling from a standing position. 

The median of the multiple trauma group’s initial hemoglobin values was 11.65 mg/L, and was slightly lower than that of the femoral fracture group, with 12.3 mg/L (*p* = 0.007). The difference, however, was less than one Hb point. The same phenomenon was observed with the lowest Hb value. The median of the lowest Hb value in the multiple trauma group was 8.1 mg/L, which was slightly lower than that of the femoral fracture group, with 8.7 mg/L (*p* = 0.007). Again, the difference was less than one Hb point. In total, 68% of the multiple trauma patients and 59% of the femoral fracture patients received a transfusion of red cell concentrates. The difference between the patient groups, however, was not statistically significant (*p* = 0.170). The transfusion quantities were compared in the patients in whom at least 1 EC was transfused. There was a statistically significant difference between the two patient groups (*p* < 0.001). An average of 8 erythrocyte concentrates (ECs) (Q1–Q3: 4–12 ECs) were transfused in the multiple trauma patients, and 2.5 ECs (Q1–Q3: 2–4 ECs) were transfused in the femoral fracture patients.

In addition, the length of stay in the intensive care unit was significantly longer for the patients in the polytrauma group than for the patients in the femur fracture group. The median length of stay in the intensive care unit was 6 days for the polytrauma group (Q1–Q3: 3–16 days), and only 2 days for the femur fracture group (Q1–Q3: 1–4 days).

The median length of stay in hospital was 15 days in the polytrauma group (Q1–Q3: 5–37 days) and 11 days in the femur fracture group (Q1–Q3: 9–14 days). Thus, the patients in the polytrauma group stayed in hospital for an average of 4 days longer (*p* = 0.033).

The proportion of patients discharged home was 35.6% in the femoral fracture group and 24.4% in the multiple trauma group. The proportion of patients discharged to a care facility was significantly higher in the femoral fracture group (31.1%) than in the multiple trauma group (12.2%). On the other hand, discharges to other hospitals were significantly more frequent in the multiple trauma group (33.2%) than in the femoral fracture group (16.7%). In total, 12.2% of the patients in the multiple trauma group and 17.0% of the femoral fracture group patients were discharged into rehabilitation.

The polytrauma group also included patients who also suffered femoral fractures as part of their multiple injuries. A total of 21 patients in the multiple injury group had femoral fractures, but only five of them had a proximal femur fracture.

### 3.2. Outcome Parameters

In this study, the hospital mortality was 16.7% in the multiple trauma group, but only 0.8% in the femoral fracture group. Thus, the hospital mortality in the polytrauma group was significantly increased, and the difference between the groups was statistically significant (*p* = 0.001). There was a statistically significant difference between the two groups concerning the mobility status at discharge (*p* < 0.001). All of the physiotherapy treatments were digitally documented. From this, the mobility status at discharge could be taken. In the multiple trauma group, 9% of the patients were mobile without aids, but at the same time, 34% of the patients were bedridden. Most of the femoral fracture group patients were restricted to aids, but this group showed a significantly lower proportion of patients who were bedridden on discharge (Figure 1).

### 3.3. Regression Analysis

Especially in geriatric polytrauma patients, the risk factors for functional capacity after trauma are not yet sufficiently known. Therefore, regression analyses were initially performed in order to investigate the relationship between mobility status as the dependent variable and its respective potential influencing factors. A simple linear regression analysis with mobility status as the dependent variable showed significant associations (*p* < 0.1) for the following parameters: patient group, age, number of secondary diagnoses, known dementia, ASA, intubation, initial Hb level, lowest Hb level, and EC amount (Table 3). With increasing age, number of secondary diagnoses, known dementia, higher ASA values, intubations performed, and transfused erythrocyte concentrates, the mobility status deteriorated. The lower the initial and the lowest Hb value, the worse the mobility status.

A multiple linear regression analysis was performed in order to analyze the relationship between mobility status as the dependent variable and the respective patient group (polytrauma vs. proximal femur fracture), taking into account the other influencing factors (confounders). In each case, those variables were introduced as influencing factors that previously produced significant results in the simple linear regression.

The multiple regression analysis showed that age, known dementia, intubation performed on admission, and the lowest Hb level had an impact on mobility status (F = 7.420, *p* < 0.0001).

Thus, the regression Equation (1) is as follows:Mobility state = 0.067age + 0.830dementia + 0.822intubation − 2.06lowest Hb value.(1)

If the patient’s age increases by 1 year, the mobility status worsens by 0.067 points on average. If dementia is known, the mobility status worsens by 0.830 points on average. If a patient was intubated before admission, the mobility status worsens by 0.822 points, and if the lowest Hb value is 1 mg/dL lower, the mobility status deteriorates by 2.06 on average. This means that whether a patient belongs to the polytrauma group or the femur fracture group does not have a statistically significant influence on the mobility status as long as other influencing factors are taken into account simultaneously. The conditions for the multiple linear regression were checked in advance by numerous statistical tests.

## 4. Discussion

Based on the comparison between the polytrauma and femur fracture groups, the characteristics of both groups can be summarized.

The patients in the polytrauma group were presumably more active and mobile in their daily lives than the patients in the femur fracture group. The mean age of the polytrauma patients was about 5 years younger than that of the femur group. The most frequent trauma mechanisms in the polytrauma group were falls from a height and traffic accidents. In contrast, 80% of the femur fracture patients sustained their injury from a fall from a standing position. The result of the present study shows comparable results that are analogous to previous studies [8].

The patients in the femur fracture group were presumably more comorbid (with a higher proportion taking coagulation-influencing drugs, and a higher proportion of patients with dementia) than those in the polytrauma group. However, this tendency could not be shown based on the number of secondary diagnoses or the ASA classification.

Immediately after the trauma and during hospitalization for acute trauma surgical treatment, the patients in the polytrauma group were more severely affected than those in the femur fracture group. The prehospital condition was generally more severe in the polytrauma patients than in the femur fracture patients. A significantly greater proportion of patients in the polytrauma group required intensive care. The length of stay in the intensive care unit and the total length of stay were longer in the polytrauma group than in the femur fracture group.

Functional capacity as a short-term outcome parameter had a high value. Both previous studies and the present study show that a significant proportion of patients survive their trauma event. For surviving patients and their families, long-term, post-traumatic quality of life is relevant, and intact functioning is an essential component of life quality. Functional capacity at discharge does not always reflect a patient’s long-term condition after a trauma event. After acute trauma surgery treatment, a patient is not always discharged home or to a nursing facility. For example, in this study, 32.3% of the polytrauma patients and 16.7% of the femur fracture patients were discharged to another hospital, and 12.2% of the polytrauma patients and 15.9% of the femur fracture patients were discharged to a rehabilitation facility. In such patients, their subsequent functional capacity, e.g., after rehabilitation, may differ from the functional capacity at discharge from an acute trauma facility. The functional capacity at discharge as a short-term outcome parameter is nevertheless of high importance, as some studies have shown that functional status at discharge acts as a predictor of long-term mortality [9,10].

The multiple regression analysis in this study showed that the difference between whether a patient belongs to the polytrauma group or the femur fracture group has no significant influence on the mobility status at discharge as long as other influencing factors are taken into account simultaneously. This result suggests that injury severity is not the most important factor in a geriatric patient’s short-term functional capacity after trauma once the patient has survived that trauma. The study also suggests that the risk factors for functional capacity as an outcome parameter are different from the risk factors for mortality. For mortality, injury severity appears to act as an important risk factor [11,12]. The functional status was not weighted. The recorded physiotherapeutic findings were objectively analyzed and grouped. The authors assumed that mobility would be significantly reduced if the injury was severe, with or without a femur fracture. In the case of a proximal femur fracture, on the other hand, mobility can also be reduced due to the fracture or due to the poor physical condition. This had no influence in this study. The overlap between the two groups—a total of five patients with proximal femoral fractures in the group of multiple trauma patients—was left deliberately. This overlap reflects reality, and the authors saw no reason to exclude these patients from the study.

The lowest Hb value was a significant risk factor for functional capacity in this study’s regression analysis. The lower a patient’s lowest Hb level, the worse the mobility status at discharge. A relationship between a lowered Hb level and functional loss has also been pointed out in some studies. For example, Zilinski et al. reported the negative effects of anemia in patients with a minimum age of 70 on the multidimensional geriatric assessment, which evaluates a patient’s functional, cognitive, and emotional capacities [13]. In trauma surgery, postoperative anemia may occur as a complication in addition to pre-existing chronic anemia. In this study’s patient population, almost 90% of the polytrauma patients and more than 95% of the patients with a femur fracture showed their lowest Hb value only during the inpatient course. In the regression analysis, the lowest Hb value was shown to be a significant risk factor for mobility status at discharge, but not the initial Hb value at admission. This suggests that the sufficient management of postoperative anemia may contribute to better functional capacity at discharge.

In this regression analysis, the following four parameters were identified as significant factors influencing mobility status at discharge: age, known dementia, intubation, and the lowest Hb value. Age and cognitive impairment have also been identified as risk factors in previous studies of geriatric patients with a femur fracture. The results of this study suggest that these factors are also relevant in polytrauma patients. However, age and dementia can hardly be influenced by the currently-available medical intervention options during regular inpatient treatment. However, in the future, these parameters could contribute to a more efficient allocation of scarce medical resources through a better assessment of the prognosis concerning functional capacity.

Certain limitations of this study are seen in the retrospective collection of the patient data.

## 5. Conclusions

Considering the possible influencing factors, the present study shows no significant influence on mobility status at discharge when comparing both patient groups. The significant factors influencing mobility status at discharge were identified as age, known dementia, prehospital intubation, and the lowest Hb level. Here, the lowest Hb value is the only risk factor that can be influenced, if necessary, during the inpatient course. Future improvement in short-term functional capacity after trauma is conceivable through further research regarding peri- and postoperative anemia management in geriatric patients.

Despite their advanced age and vulnerability, a large proportion of geriatric patients survive traumatic events. Further prospective studies on maintaining or restoring functionality after an accident are necessary. Likewise, patients’ outcomes should be studied for a longer interval after discharge from the inpatient setting in order to compensate for short-term effects.

## Figures and Tables

**Figure 1 jcm-10-01287-f001:**
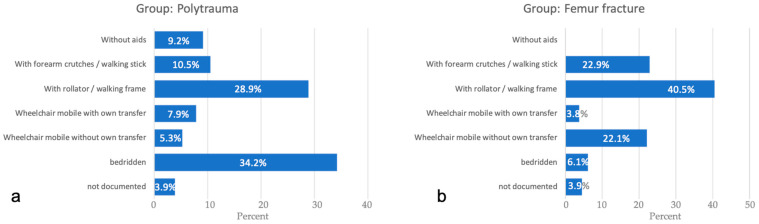
Mobility status at discharge: (**a**) polytrauma group; (**b**) femur fracture group.

**Table 1 jcm-10-01287-t001:** Collected parameters for the representation of the group characteristics.

Group of Parameters Surveyed	Detailed List of Parameters
Patient-related parameters	Age at admission; gender; number of secondary diagnoses; known dementia; Anticoagulant medication upon admission; ASA classification
Parameters for pre-clinical condition	Glasgow Coma Scale; Shock index; Intubation; resuscitation; catecholamines
Trauma-related parameter	Injury Severity Score (ISS) Trauma mechanism (fall: fall from a standing position; fall up to 1.5 m; fall up to 3 m; fall up to 6 m; fall over 6 m; Stair fall: Stair fall up to 5 steps; Stair fall over 5 steps); cycling accident; Scooter/motorcycle accident; car accident; pedestrians vs. car; others
Clinical parameters	Hemoglobin value (initial and lowest measured value); Difference between the initial and the lowest Hb value; Number of packed red blood cells transfused; stay in the intensive care unit > 24 h; length of stay in the intensive care unit
Hospital care parameters	Total length of stay in hospital; place of discharge or relocation

**Table 2 jcm-10-01287-t002:** This table shows the patients’ pre-clinical data related to intubation, resuscitation, and catecholamines.

Pre-Clinical State		Polytrauma	Femur Fracture	*p*-Value
Intubation	Yes number (%)	29 (32.2%)	0 (0.0%)	<0.001
	No number (%)	61 (67.8%)	128 (100.0%)	
Resuscitation	Yes number (%)	2 (2.2%)	0 (0.0%)	0.092
	No number (%)	89 (97.8%)	128 (100.0%)	
Catecholamine	Yes number (%)	32 (14.6%)	0 (0.0%)	<0.001
	No number (%)	59 (85.4%)	128 (100.0%)	

**Table 3 jcm-10-01287-t003:** Simple linear regression with mobility status at discharge as the dependent variable; regression coefficients with the standard error, *p*-value, and coefficient of determination(R^2^).

	Regression Coefficient	Standard Error	*p*-Value	R^2^
Patient group	−0.503	0.216	0.021	0.022
Age	0.060	0.017	0.001	0.055
Number of secondary diagnoses	−0.004	0.159	0.985	0.001
Anticoagulation	0.167	0.048	0.001	0.054
Dementia on admission	0.196	0.321	0.362	0.004
ASA	1.125	0.292	<0.001	0.065
GCS	0.628	0.219	0.005	0.041
Shock index	−0.065	0.040	0.106	0.009
Resuscitation (Yes/No)	0.086	0.504	0.865	0.001
Intubation (Yes/No)	0.358	1.501	0.812	0.001
Catecholamine administration (Yes/No)	0.912	0.346	0.009	0.030
Trauma mechanism	0.351	0.574	0.541	0.002
Initial Hb value	0.334	0.227	0.142	0.011
Lowest Hb value	−0.126	0.056	0.024	0.021
EC amount	−0.246	0.075	<0.001	0.048

Patient group (0: polytrauma, 1: femur fracture), anticoagulation (0: none, 1: taking anticoagulants), dementia on admission (0: none, 1: known dementia), ASA (0: ASA 1 or 2, 1: ASA 3 or 4), resuscitation (0: no, 1: yes), intubation (0: no, 1: yes), catecholamine administration (0: no, 1: yes), trauma mechanism (0: low-energy, 1: high-energy), EC amount: number of red blood cell concentrates transfused.

## Data Availability

The data presented in this study are available on request from the corresponding author. The data are not publicly available due to ethical principles.

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
