# Peer review of "Comparing the Short-Term Outcome after Polytrauma and Proximal Femur Fracture in Geriatric Patients"

_jcm, 2021, doi:10.3390/jcm10061287_

Round 1

Reviewer 1 Report

After having reviewed the manuscript I would like to give the following comments to the authors:

Overall, the scientific value of the manuscript is questionable for the following reasons: 

  1. It is unclear, how the functional status, the primary outcome parameter of the retrospectife study, is composed. The authors present in figure 1 the different mobility states, but it is unclear how these different states are weighted/put together in order to compare the two patient groups in in regard to their functionality.
  2. Comparing these two patient groups is questionable as they have very different characteristics to start with and certain characteristics of the polytrauma group will most certainly balance out certain characteristics of the femur fracture group or vice-versa.
  3. Sample size: the authors state that they randomly selected a specific number of patients from the patient population with femur fractures in order to reach an equal number of patients. This raises the question why in the end the femur fracture patient group consists of 132 patients but the polytrauma group of 91 patients only.
  4. Collected parameters and presentation of those parameters: Of course it is understandable that due to the retrospective nature of the study the data quality cannot be influenced and certain parameters might not be available or incomplete. Nevertheless, from my point of view, some important parameters are missing, especially a more objective parameter to define and quantify the patients‘ comorbidities. Additionally, the authors state in the materials and method section that length of hospital stay was documented as well, but no information about that can be found in the manuscript and are not taken into consideration when comparing the functionality at discharge.
  5. Unfortunately no mid- or long-term outcomes are presented and no quality of live data are presented, which is most likely due to the retrospective nature of the study and those data are most likely not available.

Author Response

Dear Reviewer,

Thank you for your review of our manuscript. We have had it revised extensively with regard to the English language and made numerous changes. We have revised the introduction and explained the research design in more detail to better explain our approach. We hope that you agree to a publication with the changes described below.

1) This point was detailed in the manuscript. The functional status is documented in detail by the treating physiotherapist at the end of the inpatient stay. This made it possible to assign them to the various groups. There was no weighting, but this point was discussed again in the discussion.

2) Thank you for that comment. The groups seem and are different. However, so far we have mostly assumed that the group of multiple trauma patients have a worse outcome. This study shows that this is the case at the end of the inpatient stay, but in the medium and long term, according to our own studies, this does not seem to be the case. It is therefore still important to publish the results of this study and to show relevant differences between these patient groups. In relation to the intersection of the two groups, it can be said that a total of 21 patients in the multiple trauma group have a femur fracture, 5 of them a proximal femur fracture. These patients were deliberately left in the study collective. We have explained and illustrated this in more detail in the manuscript.

3) The sample size was the same at the initial time point, but later patients had to be excluded from the polytrauma group whose traumatic brain injury was too severe. We decided to leave the femur fracture group larger anyway, as it ultimately makes no difference in the observation and the group size is still comparably large. We have highlighted this in more detail in the manuscript.

4) Thank you for this comment on the collected parameters. We have added both the length of intensive care stay and the duration of the hospital stay. Since the data collected were not statistically significant with regard to the length of stay, they are not directly included in the comparison of the functionality.

It is true that the simple counting of secondary diagnoses must be critically questioned. The Charlton Comorbidity Index was not used, but allows better predictive power with regard to morbidity. However, the simple counting methods of the secondary diagnoses showed a higher quality than the scores relating to hospitalization.

5) In this study, only the outcome at the end of the inpatient stay was compared, so unfortunately no data on the medium- or long-term outcome are available that can be supplemented in this manuscript. 

Reviewer 2 Report

Overall the introduction is a bit too wordy.

The goal of the study is unclear.

I couldn’t tell exactly what you intended to  compare and why:

  • Is it geriatric multitrauma patients with femur fractures vs non-multitrauma geriatric patients with just an isolated femur fracture ?
  • Is it geriatric multitrauma patients without  a femur fracture vs non multitrauma geriatric patients with just an isolated femur fracture?
  • Is if geriatric multitrauma patients with or without a femur fracture vs non multitrauma geriatric patients with just a femur fracture?

The methods sections suggest it is scenario 2) or 3) , but still not clear until later.

What is the goal:

  • If the intent is 1) as above, then is the goal to see if multitrauma makes the outcome of a femur fracture worse?
  • If the intent is 2 or 3) ) as above is the goal to see if multitrauma has a worse outcome  in a geriatric patient that just a  femur fracture?  whether  the multitrauma involved a femur fracture or not?

I would focus on making a more compact statement of why this study is necessary and the specific purpose for it.

Overall, I understand the study is really an analysis of multitrauma patients with or without a femur fracture vs non multitrauma geriatric patients with just a femur fracture. Frankly, I am not sure that this is a very useful comparison. It just two very different mechanisms. You are comparing the outcome of an isolated femur fracture, likely a low energy injury, with the outcome of some multitrauma situation which may or may not also include a femur fracture. I would focus your study on comparing outcomes of non-multitrauma femur fractures with outcomes of femur fractures where the patient has sustained multitrauma

Author Response

Dear Reviewer, 

Thank you for your review of our manuscript. We have made numerous changes. We hope that you agree to a publication with the changes described below.

1) Thank you for that comment. We have revised the introduction and in particular detailed the aim of the study.

2) We have explained this in more detail in the aims of the study and defined more clearly which patients are compared and what the goal is.

3) Many thanks for these comments. We have revised the section significantly. In summary, scenario 3 was examined. 5 patients from the group of multiple trauma sufferers also had a proximal femur fracture. These patients were deliberately left in the group.

4) Thank you for pointing this out. In fact, there are basically two quite different patient groups. Nevertheless, it is the patients that we see frequently or more frequently in everyday clinical practice and for whom the outcome is interesting. Even if the accident mechanism is very different, the outcome is less different than initially assumed. In particular, but this is not part of the present manuscript, the quality of life of the multiple trauma sufferers is better than expected. The proposed comparison of femoral fractures with and without multiple trauma is a very good consideration. Due to the group size, however, no longer adaptable for this study. Nevertheless, this study also gives us valuable information for our daily work in the care of our patients and we will follow your advice for future studies.

Round 2

Reviewer 1 Report

Thanks a lot for addressing my comments and making the respective changes in the manuscript. The manuscript has its limitations, but after the revision I can recommend the manuscript for publication.

Reviewer 2 Report

Thank you for addressing my recomendations